# Peritoneal Fluid Analysis of Advanced Ovarian Cancers after Hyperthermic Intraperitoneal Chemotherapy

**DOI:** 10.3390/ijms24119748

**Published:** 2023-06-05

**Authors:** Wei-Chun Chen, Ting-Chang Chang, Hung-Hsueh Chou, Mei-Hsiu Cheng, Jun-Jie Hong, Yi-Shan Hsieh, Chao-Min Cheng

**Affiliations:** 1Division of Gynecologic Oncology, Department of Obstetrics and Gynecology, Chang Gung Memorial Hospital at Linkou, College of Medicine, Chang Gung University, Taoyuan 333, Taiwan; lionsmanic@gmail.com (W.-C.C.); tinchang.chang@gmail.com (T.-C.C.); ma2012@cgmh.org.tw (H.-H.C.); 2Department of Obstetrics and Gynecology, New Taipei City Municipal Tucheng Hospital, New Taipei City 236, Taiwan; 3International Intercollegiate Ph.D. Program & Institute of Biomedical Engineering, National Tsing Hua University, Hsinchu 300, Taiwan; 4Institute of Biomedical Engineering, National Tsing Hua University, Hsinchu 300, Taiwan; 5School of Traditional Chinese Medicine, Chang Gung University, Taoyuan 333, Taiwan; 6School of Medicine, National Tsing Hua University, Hsinchu 300, Taiwan; 7Taiwan Business Development Department, Inti Taiwan, Inc., Hsinchu 302, Taiwan; michelle@intilabs.com (M.-H.C.); winterhong@intilabs.com (J.-J.H.); sandra@intilabs.com (Y.-S.H.)

**Keywords:** ovarian cancer, cytoreduction surgery, hyperthermic intraperitoneal chemotherapy, HIPEC, miRNA, cytokine

## Abstract

This study investigated miRNA and cytokine expression changes in peritoneal fluid samples of patients with advanced ovarian cancer (OVCA) after receiving hyperthermic intraperitoneal chemotherapy (HIPEC) during cytoreduction surgery (CRS). We collected samples prior to HIPEC, immediately after HIPEC, and 24/48/72 h after CRS from a total of 6 patients. Cytokine levels were assessed using a multiplex cytokine array, and a miRNA PanelChip Analysis System was used for miRNA detection. Following HIPEC, miR-320a-3p, and miR-663-a were found to be immediately down-regulated but increased after 24 h. Further, significant upregulation post-HIPEC and sustained increases in expression were detected in six other miRNAs, including miR-1290, miR-1972, miR-1254, miR-483-5p, miR-574-3p, and miR-574-5p. We also found significantly increased expression of cytokines, including MCP-1, IL-6, IL-6sR, TIMP-1, RANTES, and G-CSF. The changing expression pattern throughout the study duration included a negative correlation in miR-320a-3p and miR-663-a to cytokines including RANTES, TIMP-1, and IL-6 but a positive correlation in miRNAs to cytokines including MCP-1, IL-6sR, and G-CSF. Our study found miRNAs and cytokines in the peritoneal fluid of OVCA patients demonstrated different expression characteristics following CRS and HIPEC. Both changes in expression demonstrated correlations, but the role of HIPEC remains unknown, prompting the need for research in the future.

## 1. Introduction

Ovarian malignancies are one of the most common lethal cancers among women. On an annual, global basis, ovarian malignancies account for nearly 310,000 new-onset cases and approximately 200,000 deaths [1]. Most early-stage ovarian cancers initially present nonspecific symptoms such as abdominal distention, abdominal fullness, bowel habit change, or poor appetite. Consequently, 70% of ovarian cancers have been diagnosed at an advanced stage with metastasis [2]. Stage 3 or 4 ovarian cancers are prone to produce ascites [3], and the formation of ascites may be the result of peritoneal tumor metastasis leading to tumor obstruction of lymphatic drainage [4] and angiogenesis-related vascular permeability changes associated with several growth factors such as VEGF (vascular endothelial growth factor), FGF (fibroblast growth factor), and PDGF (platelet-derived growth factor) [5,6]. The accumination of ascites may cause abdominal compression symptoms leading to anorexia, fatigue, cachexia, and even dyspnea [7,8]. Such morbidities have been shown to decrease patient survival in cases reported up to 20 weeks [9,10].

The malignant ascites of ovarian cancers include a unique immunosuppressive tumor microenvironment comprising cellular and noncellular elements of ascites and tumor mesenchyme [11,12,13]. The profound levels of immune-suppressive cytokines such as vascular endothelial growth factor A (VEGF-A), interleukin 10 (IL-10), and transforming growth factor beta 1 (TGF-β1) in ascites have an immunosuppressive effect [14]. In malignant ascites of ovarian cancer patients, the lymphocyte composition showed higher expression of CD4 and CD8 T cells relative to immunosuppressive receptors such as LAG-3: lymphocyte-activation gene-3 (LAG-3), programmed death-1 (PD-1), T cell immunoglobulin and mucin-domain-containing-3 (TIM-3), and cytotoxic T-lymphocyte-associated protein 4 (CTLA-4), particularly when compared to peripheral blood samples (15). Further, these immunosuppressive receptors on tumor-associated T-cells in the malignant ascites of ovarian cancer patients can also induce cellular immunosuppressive signal transduction such as NF-k and NFAT [15]. Previous literature has discussed several ascites biomarkers for ovarian cancer. Patients with more tumor necrosis factor alpha (TNF-α), IL-6, osteoprotegerin (OPG), IL-10, and leptin in ascites had a poorer prognosis, a shorter PFS, and more resistance to chemotherapy [16,17]. More interferon gamma (IFN-γ) in ascites may be related to stronger immune escape and, therefore, a more advanced disease stage, poorer survival, and more difficulty in achieving complete cytoreduction [18,19].

MicroRNAs (miRNAs) are endogenous non-coding RNAs secreted by certain cells, including malignant tumor cells [20]. These molecules undergo intracellular processing to form RNA silencing complexes (RISC) that regulate gene expression by degrading targeted mRNA or interfering with protein synthesis [21]. MiRNAs are implicated in tumorigenesis through genetic alterations and have been identified as potential biomarkers for differentiating ovarian cancer from benign tumors [22,23]. They are also associated with prognosis and survival in ovarian cancer treatment. miRNA microarrays studies such as customized TaqMan low-density miRNA arrays can screen the expression levels of 48 miRNAs in sera from epithelial ovarian cancer (EOC) patients and benign ovarian tumor patients and further found miRNA-20a, miRNA-125b, miRNA-126, and miRNA-355 had significant differences [24]. Another study also developed a 35 miRNA-based classifier, MiRO-vaRv, to evaluate the miRNA expression profile of EOC patients [25]. Due to their detectability in various body fluids and diverse expression profiles, miRNAs are promising biomarkers for cancer research.

Hyperthermic intraperitoneal chemotherapy (HIPEC) performed during CRS has been found to provide a significant survival benefit for primary advanced ovarian cancers after neoadjuvant chemotherapy as well as ovarian cancers with multiple relapses [26,27]. HIPEC is especially good at treating intraabdominal lesions with heat and intraperitoneal chemotherapy. While the ascites profile of cytokines and miRNAs may be affected, the related issues have scarcely been discussed. Therefore, the aim of our present study is to investigate varying expressions before and after HIPEC treatment for ovarian cancer and evaluate the potential effects and reactions of HIPEC treatment.

## 2. Results

### 2.1. Patient Characteristics

In our study, peritoneal fluid samples from six patients were collected between January 2022 and May 2022 in a single tertiary hospital. Patient characteristics, demonstrated in Table 1 and Figure 1, reveal the overall study scheme protocol. Among the enrolled patients, ages ranged from 49 to 66 years of age, the body matrix index was approximately 16.63 to 29.61, all International Federation of Gynecology and Obstetrics (FIGO) stages were at least stage 3 or 4, and the histology was high-grade serous carcinoma in 5 cases, with 1 case graded as low-grade serous carcinoma. Three patients received 3 to 4 courses of neoadjuvant chemotherapy containing carboplatin and paclitaxel before CRS and HIPEC. The peritoneal carcinomatosis index (PCI) measures the degree of abdominal cancer dissemination from 0 (absence of disease) to 39 (widespread disease), accounting for tumor size and spread across 13 peritoneal areas. The completeness of the cytoreduction (CC) score rates the effectiveness of cytoreductive surgery, from CC-0 (no remaining visible disease) to CC-3 (non-resectable tumor nodules). Both scores are essential for assessing peritoneal cancer conditions and the outcomes of surgical intervention [28,29]. During CRS, the PCI score ranged from 5 to 16. One case had a CC score of 1, and the other cases had complete cytoreduction status. We collected samples from 1 case with CRS alone, and the other cases had CRS plus HIPEC using carboplatin plus paclitaxel. Five cases had no evidence of disease after primary treatment, but one case had disease progression with a PFS of 8.57 months. The detailed data is listed in the Appendix A.

The study scheme is illustrated in Figure 1. The samples from cases A, B, C, and D were collected and used for cytokine microarray evaluation. In addition, samples from the other cases (patient E had HIPEC for low-grade serous ovarian cancer, and patient F did not undergo HIPEC) were investigated for miRNA analysis and further ontology enrichment.

### 2.2. miRNA Expression before and after HIPEC

Table 2 shows the comparison in miRNA expression among the peritoneal fluid samples between pre-HIPEC and immediately post-HIPEC status in patient E after CRS. There were 8 miRNAs with significantly differential expression, including 6 with increasing and 2 with decreasing expression following HIPEC treatment. In our study, a “|ΔΔCq|” value over 1 was identified as an miRNA value with significantly differential expression. There were 6 miRNAs, including miR-1254, miR-1290, miR-1971, miR-483-5p, miR-574-3p, and miR-574-5p, with expression differences, and the ΔΔCq levels were 4.91, 2.85, 3.76, 2.73, 6.26, and 2.27, respectively. Two miRNAs, including miR-320a-3p and miR-663a, with ΔΔCq levels of approximately −5.31 and −5.85, respectively, were detected with significantly decreased expression following HIPEC treatment.

We identified specific miRNAs exhibiting significant expression differences in samples pre- and post-HIPEC treatment. Figure 2 shows the changing pattern of miRNA expression from pre-HIPEC status to 3 days after HIPEC treatment. Both miR-320a-3p and miR-663a demonstrated decreased expression immediately after HIPEC treatment, but expression steadily increased starting at 24 h after HIPEC treatment. The other 6 miRNAs listed above had significantly increased expression after HIPEC treatment, and expression remained significantly increased at 24, 48, and 72 hr. However, expression at 72 h post-HIPEC treatment had decreased below peak expression levels. There was no difference in expression between the patient with stage 4B cancer (A) and the patients with stage 3C2 cancer (B, C, and D).

### 2.3. miRNA Expression between CRS with or without HIPEC

To understand the impact of the differences between CRS and HIPEC on patient abdominal fluid miRNA profiles, we also studied miRNA expression in peritoneal fluid samples from patients E and F with CRS plus HIPEC and CRS alone, respectively, and the related results are listed in Table 3. In our study, there were a total of 13 miRNAs detected with |ΔΔCq| >1, signifying significantly differential expression in patients with CRS plus HIPEC treatment. Among these miRNAs, 8 miRNAs, including miR-1290, miR-596, miR-191-5p, miR-320a-3p, miR-885-5p, miR-1228-5p, miR-589-5p, and miR-202-3p, demonstrated significantly increased expression. Relatively speaking, the other 5 miRNAs involving miR-1972, miR-423-5p, miR-30b-5p, miR-151a-5p, and miR-378a-5p showed significantly decreased expression in those patients with CRS plus HIPEC compared to those with CRS alone.

### 2.4. Cytokine Expression before and after CRS Plus HIPEC

We also used a cytokine array to investigate the expression differences in cytokine profiles in peritoneal fluid over time, from pre-HIPEC status to 3 days post-HIPEC treatment, among the samples from patients A, B, C, and D. As shown in Appendix A, significantly differential expression between pre-HIPEC status and the immediate post-HIPEC samples was found in six cytokines: Regulated on Activation, Normal T-cell Expressed and Secreted (RANTES, also known as C-C motif chemokine ligand 5 or CCL5), granulocyte colony stimulating factor (G-CSF), chemokine ligand 5 (CCL2, also known as monocyte—chemoattractant protein—1 or MCP-1), tissue inhibitor of metalloproteinase-1 (TIMP-1), interleukin-6 soluble receptor (IL-6sR), and IL-6. The differences in the expression of these cytokine levels are also illustrated in Figure 3. There was no difference in expression between the patient with stage 4B cancer (A) and the patients with stage 3C2 cancer (B, C, and D).

Figure 4 demonstrates the changing cytokine expression pattern over time, with significantly differential expression after HIPEC when the observational interval spanned from a point just before HIPEC to 72 h post-HIPEC. We observed that periotineal G-CSF, IL-6, RANTES, and TIMP-1 levels peaked post-HIPEC or 24 h after HIPEC treatment and then decreased 48 h after HIPEC treatment. CCL2 was detected at significantly increased levels after HIPEC treatment, and slightly elevated levels in a steady pattern were also found at the following detection timing. IL-6sR was found with an increasing pattern immediately post-HIPEC, 48 h, and 72 h after HIPEC treatment, but there was a decreased level found at 24 h post-HIPEC treatment.

### 2.5. A Comparison of Changes in Expression Pattern between Cytokine and miRNA before and after HIPEC

The correlation and comparison between the changing expression pattern of miRNA and cytokines during the observation time from pre-HIPEC until 72 h after HIPEC treatment were analyzed using a Pearson correlation and the results are illustrated in Figure 5. The color of each grid reveals the correlation values between each cytokine and miRNA. The correlation “0” is illustrated as a yellow color; the red color indicates a positive correlation over 0.5; and blue represents a negative correlation of −0.5. Additionally, the darker shades of red and blue indicate larger correlation values.

The changing MCP-1 pattern had the strongest positive correlation with miR-1254 (*p* = 0.0269), and other detected miRNAs, including miR-574-5p, miR-483-5p, miR-574-3p, miR-1972, and miR-1290, also had positive correlations over 0.7. IL-6sR and G-CSF had the strongest positive correlations with miR-483-5p (*p* = 0.0285) and miR-1290 (*p* = 0.1552), respectively. Both RANTES and TIMP-1 had the strongest negative correlation with miR-663a (*p* = 0.0757 and 0.0192, respectively), and a negative correlation (over −0.7) was also detected with miR-320a-3p. IL-6 had the strongest correlation with miR-320a-3p (*p* = 0.0193), and miR-663-a was found to have a negative correlation of over −0.7. The relationship with the strongest correlation level between the cytokines and the miRNAs is illustrated as a Pearson correlation coefficient in Figure 6. A strong correlation with an r value over 0.7 can be found in the demonstrated comparisons.

### 2.6. Biological Process Enrichment Gene Ontology (GO) Terms

The biological process enrichment revealed that our miRNAs with significantly differential expression participated in several functional groups. Figure 7 lists the top 10 biological processes in GO terms, including epithelial cell migration, ameboidal-type cell migration, epithelium migration, and tissue migration. Table 4 lists the targeted genes involved in individual GO term groups that were regulated by our obtained miRNAs. There were many genes that simultaneously participated in several functional groups, such as the TGFB1 gene, which was involved in epithelial cell migration, positive regulation of angiogenesis, leukocyte proliferation, and response to oxygen levels. Therefore, the above biological GO functional group can form a network based on the significance of relationships between the groups and involved genes. The network of clustered gene sets and group networks, illustrated as an enrichment map in Appendix A, can present an easier method to identify the enrichment results.

## 3. Discussion

Advanced ovarian cancer is notorious for the easy formation of ascites, leading to compression symptoms and high recurrence rates post-primary treatment. To address this, multimodal anticancer treatments, including hyperthermic intraperitoneal chemotherapy (HIPEC), have been developed. HIPEC employs chemotherapy and hyperthermic perfusions administered intraabdominally at 42–43 °C, effectively targeting intraperitoneal dissemination—the primary route of advanced ovarian cancer spread. Existing literature highlights that malignant ascites and ovarian cancer can create a unique intraabdominal tumor microenvironment, characterized by anti-tumor immunity inhibition and heightened levels of immunosuppressive cytokines such as VEGF-A, IL-10, TGF-β1, and immune cells expressing immunosuppressive receptors such as LAG-3, PD-1, TIM, and CTLA-4 [14,30]. Furthermore, increased expression of proteins related to cellular proliferation pathways, such as AKT (also known as protein kinase B), ERK (extracellular signal-regulated kinase), CREB (cAMP-responsive element binding protein), and c-JNKs (c-Jun N-terminal kinases), has been reported in malignant ascites, detected via protein lysate microarrays [31]. Our study is motivated by the potential impact of HIPEC on these specific cytokines or biomarkers within the unique tumor microenvironment of ovarian cancer.

miRNAs can be detected in ovarian cancers, and related malignant ascites are reported in the literature [23]. Research has shown that disease diagnosis, treatment prognosis prediction, and disease status can be traced back to miRNA, the post-transcriptional regulation of which can affect biological activities such as cell proliferation, apoptosis, or differentiation inside or outside of cells [32,33,34,35]. Thus, miRNA may be used as an innovative biomarker for cancer diagnosis, treatment approach, and treatment evaluation [32,33,34,35]. Previous research reported increased expression of miRNAs in ovarian cancer and related malignant ascites, including miR-200b-3p, miR-135b-5p, and miR-182-5p [23]. Decreased expression of miRNAs, such as miR-451a, in malignant ascites may indicate a potential tumor suppressor gene for ovarian cancer [23]. Other research has shown that significant downregulation of miR-199a-3p, miR-199b-3p, miR-199a-5p, miR-126-3p, and miR-145-5p can be found in ovarian cancer ascites-derived spheroids, which may be related to ovarian cancer progression [36]. When comparing ovarian cancer ascites miRNAs with serum miRNAs in non-cancerous cases, a strongly increased expression of the miR-200 family and miR-1290 in nearly all ovarian cancer ascites samples was detected with weaker expression of miR-30a-5p, and lower expression of miR-200b was related to longer survival [37].

miR-320a-3p, a member of the miR-320a family, was found to be associated with tumor suppressor function in many types of cancer, including gastric cancer and lung cancer [38,39]. Additionally, one previous report noted the use of miR-320a-3p delivery via gold nanoparticles as a means of targeting Sp1 in lung cancer [40]. In our study, hsa-miR-320a-3p was found to have significantly decreased expression after HIPEC treatment but soon increased after 24 h. In addition, those receiving CRS plus HIPEC treatment demonstrated increased expression of hsa-miR-320a compared to CRS alone. It appears that HIPEC may increase miR-320-3a, and the increase may start 24 h after HIPEC treatment.

Both miR-1290 and miR-1972 had significantly increased expression after HIPEC treatment, and the increase remained after 24 h until 48 and 72 h after HIPEC treatment, respectively. However, those receiving HIPEC plus CRS demonstrated increased miR-1290 and less increased miR-1972 compared to those receiving CRS alone. In the previous literature, miR-1290 was found in ovarian cancer ascites [37], and favorable outcomes were associated with patients having high plasma levels of miR-1290 before surgery [41], but the role of miR-1290 remains unclear. One article reported that miR-1972 may have an oncogenic role in ovarian cancer, as its expression is related to cisplatin-resistant ovarian cancers, and inhibition of miR-1972 can inhibit the proliferation of cisplatin-resistant ovarian cancer cells that enhance cisplatin sensitivity [42]. Generally speaking, the roles of miR-1972 and miR-1290 are not yet confirmed, but both have demonstrated significantly differential expression following HIPEC treatment.

The other four miRNAs, miR-1254, miR-483-5p, miR-574-3p, and miR-574-5p, demonstrated significantly increased expression after HIPEC treatment, but there were no differences between CRS plus HIPEC or CRS alone. The change patterns for these 4 miRNAs were similar, i.e., increased expression can be seen immediately after HIPEC treatment; enhanced expression lasted for 48 h after HIPEC treatment and started to turn to reduction at 72 h after HIPEC treatment. Previous research has shown that miR-1254 overexpression may suppress tumor cell activity, including proliferation, migration, and invasion, further inducing cell apoptosis in glioma, gastric cancer, and oral squamous cell carcinoma [43,44,45]. Inhibition of miR-1254 could promote the activity of cervical carcinoma cells [46]. However, there was also a report showing that miR-1254 could promote tumor proliferation and invasion in hepatocellular carcinoma [47]. Previous research showed that miR-483-5p had a prometastatic function that downregulated tumor suppressors RhoGDI1 and ALCAM (activated leukocyte cell adhesion molecule) [48], inhibited miR-483-5p, and could also impede tumor proliferation in triple-negative breast cancer [49]. miR-574-3p can suppress target genes, as RAC1 and EP300 both identified its role in stimulating VEGF-mediated angiogenesis, and genistein used to treat prostate cancer can upregulate miRNA-574-3p and thus be used to treat cancer [50]. However, miRNA-574-5p has been shown to promote tumor metastasis in non-small cell lung cancer, and inhibition of miRNA-574-5p can suppress cell growth in cervical cancer cells [51]. The study reveals that miR-1254, miR-483-5p, miR-574-3p, and miR-574-5p significantly increase post-HIPEC treatment, peaking at 48 h and decreasing at 72 h. These miRNAs, with diverse roles in tumor behavior, suggest potential implications for cancer treatment.

The miRNA-663a in our study demonstrated significantly decreased expression after HIPEC treatment, and no difference was detected between those treated with CRS plus HIPEC or CRS alone. The expression of miRNA-663a decreased immediately after CRS but soon increased 24 h after HIPEC treatment. In previous research, the role of miR-663a was found to either inhibit or promote tumor cell proliferation depending on status. miR-663a demonstrated oncogenic activity to increase tumor proliferation in nasopharyngeal carcinoma [52], but other research has shown that it can act as a tumor suppressor for gastric cancer [53]. For ovarian cancers, previous research also found that miR-663a can facilitate tumor growth and invasion, and upregulated expression of miR-663a can be found in chemotherapy-resistant ovarian cancers with resultant lower survival outcomes [54,55]. In our study, generally decreased expression of miR-663a can be found after HIPEC, especially in the short term immediately after HIPEC.

Cytokines in malignant ascites, potentially linked to tumor cell immune-evasion mechanisms, could serve as prognostic indicators for patient outcomes post-cancer treatment. Markers such as TNF-α, IL-6, and IFN-γ may be tied to poor patient survival, chemotherapy resistance, and incomplete cytoreduction (18). Ascites cells can generate immunosuppressive cytokines and induce apoptosis in immune cells, fostering an immunocompromised tumor microenvironment (18, 59–61). In our study, RANTES, G-CSF, MCP-1, TIMP-1, IL-6, and IL-6sR demonstrated significantly increased expression after CRS plus HIPEC treatment. However, decreased expression after 24 or 48 h was found in G-CSF, IL-6, RANTES, and TIMP-1. Previous research found higher RANTES concentrations in the peritoneal fluid and plasma of ovarian cancer patients compared to those with benign ovarian tumors [56]. Therefore, RANTES can be used to differentiate benign from malignant ovarian tumors [57]. In addition, RANTES and proinflammatory cytokines such as IL-6 can produce a more aggressive phenotype in breast cancer cells via AKT or STAT3 signal pathways [58,59]. Previous research also reported that G-CSF may enhance tumor cell migration and impede chemotherapy-induced apoptosis in ovarian cancer cells via the JAK2/STAT3 pathway [60]. TIMP-1 is an inhibitor of the matrix metalloproteinases (MMPs) that can promote tumor cell invasion via its proteolytic enzymes for extracellular matrix degradation [61], and TIMP-1 can also lead to increased cancer-associated fibroblasts (CAFs) in prostate and colon cancer tissues and therefore facilitate cancer progression by ERK1/2 kinase activation [62]. Increased TIMP-1 levels have been related to poor prognosis in triple-negative breast cancers [63], but no evidence of a correlation with ovarian cancer patients was found [64]. In our study, the impact of HIPEC on these tumor-promoting cytokines likely started 24 or 48 h after HIPEC treatment. MCP-1 had a stable, high expression after HIPEC treatment during the observation duration, and IL-6sR demonstrated increased expression during observation after a short decrease at 24 h following HIPEC treatment. Previous research found that MCP-1 can facilitate tumor cell migration and omental metastasis of ovarian cancers via the PI3K/AKT/mTOR pathway and downstream HIF-1α and VEGF-A [65]. MCP-1 can also promote the invasion and adhesion of ovarian cancer cells [66]. Interestingly, there was also a study showing that MCP1 can be upregulated in ovarian cancers after chemotherapy, but the mechanism remains unknown [67]. In our study, we also detected such findings after HIPEC treatment, and neither a stress-induced response nor other reasons were confirmed as the cause; thus, additional research is warranted. IL-6 is a pro-inflammatory cytokine produced by tumor cells and tumor-infiltrating immune cells [68], and therefore increased IL-6 expression indicates local tumor status. IL-6 and IL-6sR were related to tumor stage and metastasis, and previous research has demonstrated that pre-operative IL-6 and IL-6sR levels could be related to a poorer prognosis for bladder cancers receiving radical cystectomy [69]. In our study, decreased IL-6 and increased IL-6sR were found at 24 and 48 h after HIPEC treatment, and the rationale for this change remains unknown and warrants further study.

Both miRNA and cytokine expression change over time from pre-HIPEC status until 72 h after HIPEC, the end of our study observation. In our study, different cytokines demonstrated positive or negative correlations with changing miRNA expression in peritoneal fluid over time, and some correlations may have differential significance. We also reviewed the downstream genes of the miRNAs with significantly differential expressions.

From the miRNA enrichment of biological processes in the gene ontology (GO), the obtained miRNAs with significantly differential expression may be involved in several functional groups, including epithelial cell migration, positive regulation of angiogenesis, leukocyte proliferation, and response to oxygen levels. Among these functional groups, there are many targeted genes such as AKT3, CD274, PTEN, SRF, SMAD4, TGFB1, CEBPB, and TP53 regulated by our detected miRNAs. Although the miRNAs identified in our study had no direct control over the genes of the cytokines with significantly differential expression by the cytokine array, these miRNAs may indirectly affect the pathways or functions involved in the genes of these cytokines, and therefore the correlated changes can be detected.

Previous research reported that the expression of hsa-miR-320a was related to an increase in inflammatory cytokines such as IL-6, MCP-1, or TNF-α, and may inhibit human-derived endothelium cell proliferation and induce apoptosis that may promote atherogenesis in coronary artery disease [70]. In our study, miR-320a expression in the intraperitoneal fluid of ovarian cancer patients had a negative correlation with RANTES, TIMP-1, and IL-6. This seems reasonable since miR-320a has demonstrated tumor suppressor activity and these 3 cytokines have relatively little oncogenic ability, but the internal relationship may need further research. Previous research has shown that down-regulation of miR-663a can also enhance cytokine secretion of TNFα, IL-1β, and IL-6 [71], a finding that was also noted in our study. miR-483-5p has also been found to have a carcinogenic role in triple-negative breast cancer in that it inhibits SCOS3 (suppressor of cytokine signaling 3) gene expression that may trigger the pathway of STAT3, NF-κB, and primary inflammatory cytokines such as TNF-α, IL-6, MCP-1, and IL-1β [49]. In our study, miR-483-5p expression also had a positive correlation with MCP-1 and IL-6sR, which may also indicate that the oncogenic role of miR-483-5p may be related to these pro-inflammatory cytokines.

To the best of our knowledge, our study is the first to investigate both miRNA and cytokine changes in peritoneal fluid samples before and after HIPEC treatment. However, we encountered some limitations. Since the peritoneal fluid sample collection was mainly done via postoperative drainage tubes, these are rarely placed for more than 3–7 days. Consequently, we only collected samples until 72 h after CRS. Since HIPEC treatment was usually performed once during the treatment course during CRS, longer-term, post-HIPEC surveillance samples may be needed to demonstrate the general change pattern among biomarkers, and such an effort should be considered for further study. In addition, we only used samples from a small number of patients, and the sample size may not be sufficient for generating consistent data among cohorts with advanced ovarian cancer. Based on our preliminary observation study, additional sample collection will be performed for future data validation.

## 4. Materials and Methods

### 4.1. Patients Collection

The samples were all collected from patients with primary ovarian cancer receiving CRS in the Linkou branch of Chang Gung Memorial Hospital. The inclusion criteria for this study were as follows: (1) patients were required to sign the informed consent; (2) patients must be diagnosed with clinical stage 3 or 4 ovarian cancers; (3) patients should receive CRS, including hysterectomy, bilateral salpingo-oopherectomy, bilateral pelvic lymph node dissection, omentectomy, and excision of all suspected tumor implants in their primary treatment. Additionally, HIPEC may be performed during surgery, but for those receiving HIPEC during CRS, the absence of extraperitoneal spreading should be confirmed before CRS, and the residual lesions after CRS should be reduced to less than 0.25 cm if possible. Beyond the above criteria, the general status of patients during CRS should be stable enough for HIPEC treatment. Those who did not meet the above criteria were not enrolled in our study.

The related clinicopathological information, including diagnostic age, body mass index, disease stage, pathology type, HIPEC regimen, tumor marker surveillance, and treatment response, was all recorded using our electronic medical chart system. The current study was conducted with the approval of the institutional review board (IRB) at Chang Gung Memorial Hospital (IRB number: 202101635B0C101).

### 4.2. CRS and HIPEC Treatment

All enrolled patients underwent CRS, including hysterectomy, bilateral adnexectomy, infracolic omentectomy, and excision of all visible intraperitoneal lesions. For those receiving HIPEC after CRS, we used a closed-system machine (Performer HT, Rand, Italy) equipped with a heating source and tubing system that connected the HIPEC machine with the intraabdominal cavity and provided for the circulation of perfusion solution inside the tubes at adjustable flow rates. During CRS, both the peritoneal carcinomatosis index (PCI) score and the completeness of cytoreduction (CC) score were recorded, and the scoring principles were assessed compared to previous articles [28,29]. In our study, the chemotherapy regimens for HIPEC were paclitaxel (135 or 175 mg/m^2^) plus cisplatin (75 or 90 mg/m^2^) or carboplatin (AUC 5 or 6). The perfusion solution for chemotherapy circulation during HIPEC used isotonic peritoneal dialysis fluid. After the tubing system was established, we added a total of 6 L of perfusion solution into circulation, including intraabdominal circulation that was kept at a fixed amount of 2 L × body surface area (BSA) for abdominal cavity distention to ensure smooth circulation without the obstruction of the tubing system by the bowel or any other intraabdominal organs. During HIPEC, we kept the temperature of the circulation perfusion between 42 and 43 °C, and the perfusion duration was 90 min for the regimen of platinum/paclitaxel combinations. After HIPEC, all of the intraabdominal perfusion solutions was evacuated using a rinsing and emptying procedure with normal saline fluid.

### 4.3. Samples Collection

We collected specimens of intraabdominal ascites or fluid (~20 mL) at the beginning of CRS. We also obtained peritoneal fluid samples following HIPEC treatment at the end of surgery and before abdominal wall closure. All of our patients receiving CRS had bilateral drainage tubes implanted at the low abdomen for fluid drainage and postoperative condition monitoring. We were subsequently able to collect peritoneal fluid samples at 24 h, 48 h, and 72 h following CRS. Following bedside sample collection, these peritoneal fluid samples were immediately sent to the laboratory and frozen at −80 °C until further examination. Drainage tubes were usually removed 3 to 5 days following CRS when the drainage amount was clean and decreased without abnormal signs of internal bleeding, intraabdominal infection, or intraabdominal organ perforation during postoperative surveillance. The study scheme is illustrated in Figure 1.

### 4.4. Cytokine Microarray Analysis

The ascites protein detection in our study was investigated using a multiple protein microarray (multiplex cytokine array, RayBiotech Inc., Peachtree Corners, GA, USA). The multiple protein microarray employed multiple enzymes combined with an immunosorbent assay (multiplex ELISA) that could simultaneously measure more than 40 different proteins, including cytokines, on a single glass chip [16,72]. Instead of directly providing the levels of the detected proteins, the cytokine microarray demonstrated the measured results by fluorescence expression, and the relative fluorescent assessment indicated the relative signal intensity that could be captured and quantified by a dual-color confocal laser scanner.

After the scanner capture, further data processing was undertaken with TiGR_Spotfinder software to interpret the stored image files after subtracting the local background interference value, so that the precise relative signal intensity of the fluorescence expression from the assessed microarray could be obtained. Additionally, the internal negative control was used to determine the threshold intensity of the positive signal. Using the signal intensity of the microarray, we could obtain the relative fluorescence expression value between different peritoneal fluid samples and thus detect targeted proteins or cytokines with a significant expression difference between the different samples tested.

### 4.5. miRNA Analysis

The collected peritoneal fluid samples were also used to detect the different types and amounts of miRNA using MIRAscan and NextAmp™ Analysis System. MIRAscan (Inti Taiwan, Inc., Hsinchu, Taiwan), is a miRNA detection assay run on NextAmp™ Analysis System (Quark Biosciences Taiwan, Inc., Hsinchu, Taiwan). MIRAscan utilizes real-time quantitative PCR (*polymerase chain reaction*) to quantify the gene expression on NextAmp™ Analysis System’s PanelChip^®^ technology. The PanelChip^®^ is a small chip with a size of 36 mm × 36 mm × 1 mm, and every single chip is comprised of 2500 nanowells that could accommodate one real-time PCR in each well. We could, thus, perform multi-gene PCR on a single chip using the PanelChip^®^ platform [73]. As with the MIRAscan assay, it contains 83 different miRNAs that could be used for different diseases and status measurements.

To perform the miRNA analysis, the sample was added to the MIRAscan PanelChip^®^ for a multi-gene qPCR reaction, and the raw Cq values representing the miRNA levels were generated for subsequent analysis. Further processing was done to eliminate miRNAs without amplification signals and normalize them with internal control; these normalized miRNA expression levels were called ∆Cq. The comparison of miRNA expression levels between the two groups can be seen from the relative change of Cq value (∆Cq), and the miRNA with significant expression differences for each comparison is identified as “|∆∆Cq| ≥ 1”. The miRNA with significantly differential expression could be used for further microRNA target interaction (MTI) analysis by miRTarBase, which is a large miRNA database containing evidence-based and experimentally validated information from numerous published articles. From the miRTarBase processing, the MTI with strong experimental evidence such as qPCR, a reporter assay, and western blot were included in subsequent analysis. The obtained MTIs could then be used for gene set enrichment analysis and functional analysis by clusterProfiler. Through this detection platform, we obtained specified miRNAs with significant expression differences between samples before and after HIPEC treatment. The miRNA detection between the samples from patients receiving CRS and those who did not receive HIPEC treatment can also be investigated. The above results can be used to observe the impact of HIPEC on miRNA expression in the intrabdominal fluid.

### 4.6. Statistics

We used SPSS (version 22.0, IBM, Chicago, IL, USA) for data analysis. To compare the cytokine levels before and after CRS with or without HIPEC, we used the paired t test to detect the cytokines with significant differences. To investigate the expression level pattern of the miRNA as well as cytokines with significantly differential expression from pre-HIPEC status until 72 h after HIPEC treatment, we used the Pearson correlation coefficient for analysis. Additionally, while the *p*-value was less than 0.05, the analyzed results were considered significant.

## 5. Conclusions

We noted that changes in miRNA expression, including miR-320a-3p, miR-663-a, miR-1972, miR-1290, miR-1254, miR-574-3p, miR-574-5p, and miR-483-5p, could be detected in the peritoneal fluid of patients with ovarian cancer after receiving HIPEC treatment. We also found changes in cytokine levels, notably MCP-1, IL-6, IL-6sR, TIMP-1, RANTES, and G-CSF. We also observed a correlation between the changing pattern of miRNA and that of cytokines. Similar negative correlations were found in miR-320a-3p and miR-663-a with cytokines such as RANTES, TIMP-1, and IL-6, but other miRNAs had positive correlations with MCP-1, IL-6sR, and G-CSF. The role of these correlated miRNAs and cytokines in HIPEC treatment needs further study and validation and should be the target of future research.

## Figures and Tables

**Figure 1 ijms-24-09748-f001:**
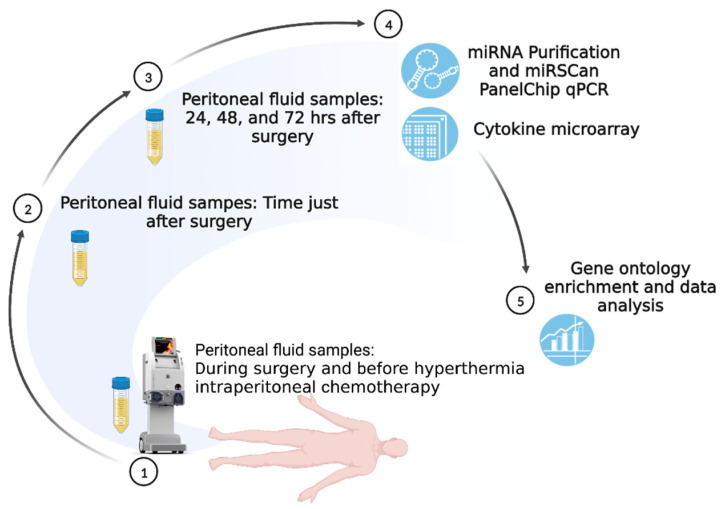
Study Scheme.

**Figure 2 ijms-24-09748-f002:**
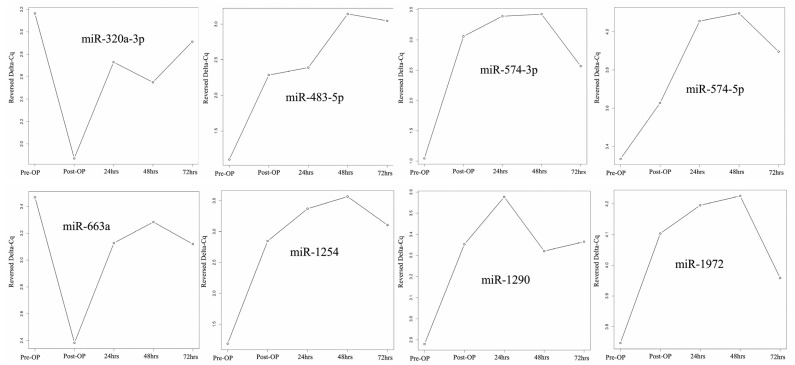
miRNA expression change from pre-HIPEC to 72 h post-HIPEC (patients A to D).

**Figure 3 ijms-24-09748-f003:**
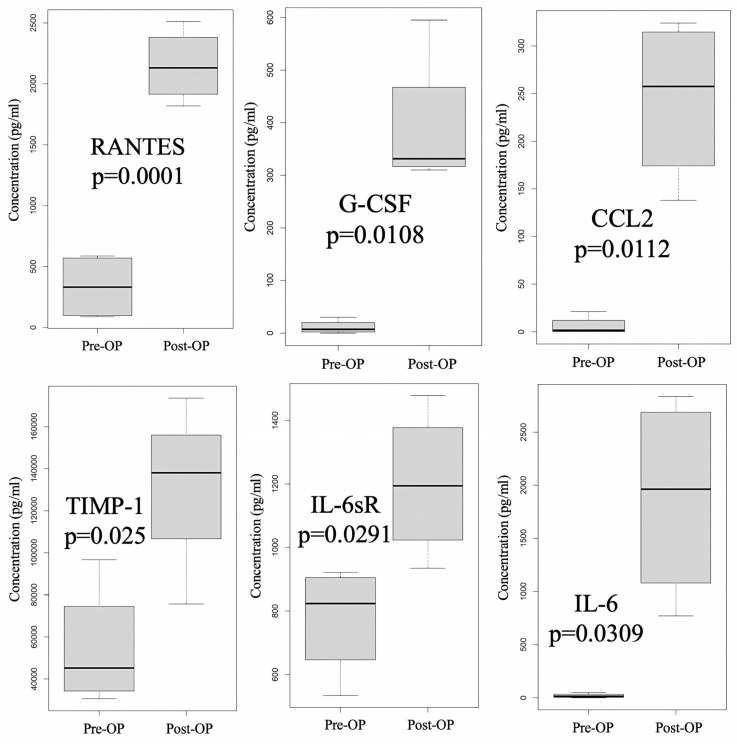
Cytokine with significantly differential expression before and after HIPEC (patients A to D). Significant differential expression in six cytokines—RANTES (Regulated on Activation, Normal T-cell Expressed and Secreted, also known as CCL5), G-CSF (Granulocyte Colony Stimulating Factor), CCL2 (also known as MCP-1), TIMP-1 (Tissue Inhibitor of Metalloproteinase-1), IL-6sR (Interleukin-6 Soluble Receptor), and IL-6—between the pre- and immediate post-HIPEC stages was identified by using a cytokine array.

**Figure 4 ijms-24-09748-f004:**
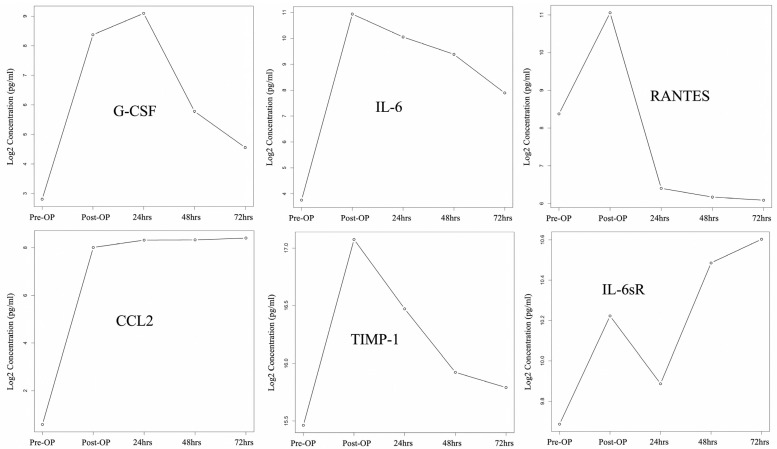
Cytokine expression changes from pre-HIPEC to 72 h post-HIPEC (patient A to D).

**Figure 5 ijms-24-09748-f005:**
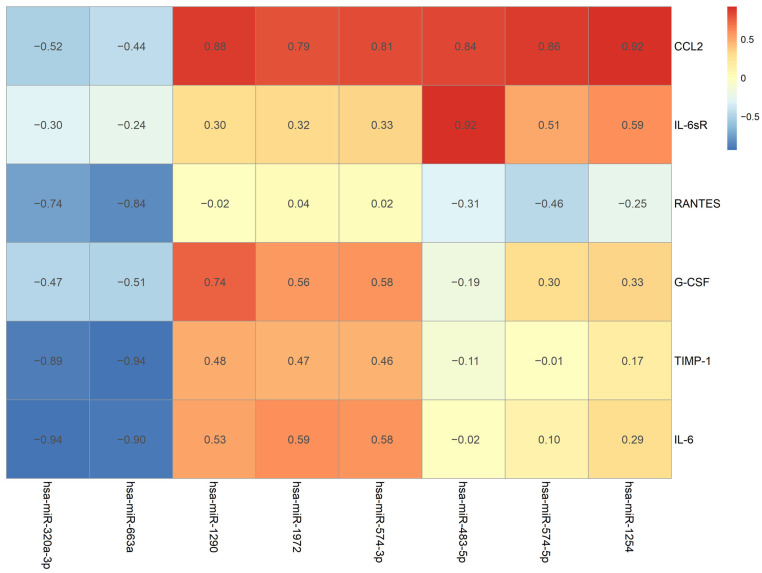
The change pattern correlation between cytokine and miRNA from pre-HIPEC to 72 h after HIPEC.

**Figure 6 ijms-24-09748-f006:**
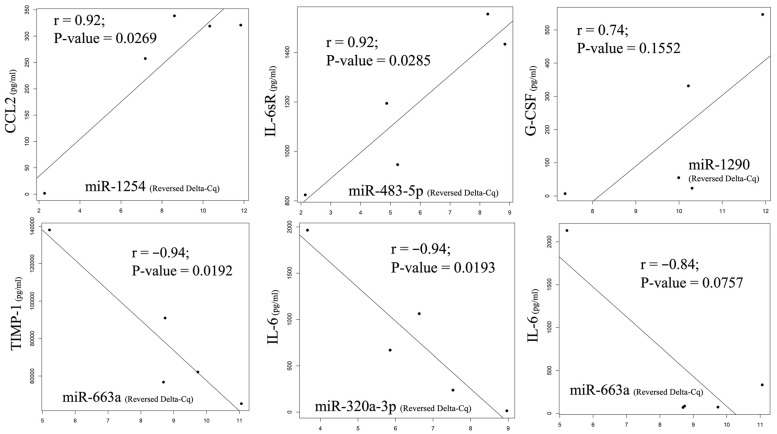
The change pattern correlation between cytokine and miRNA from pre-HIPEC to 72 h after HIPEC by Pearson correlation coefficient. A strong positive correlation was found between MCP-1 and miR-1254 and between IL-6sR and miR-483-5p, as well as G-CSF and miR-1290. On the other hand, negative correlations were observed between RANTES, TIMP-1, IL-6, miR-663a, and miR-320a-3p.

**Figure 7 ijms-24-09748-f007:**
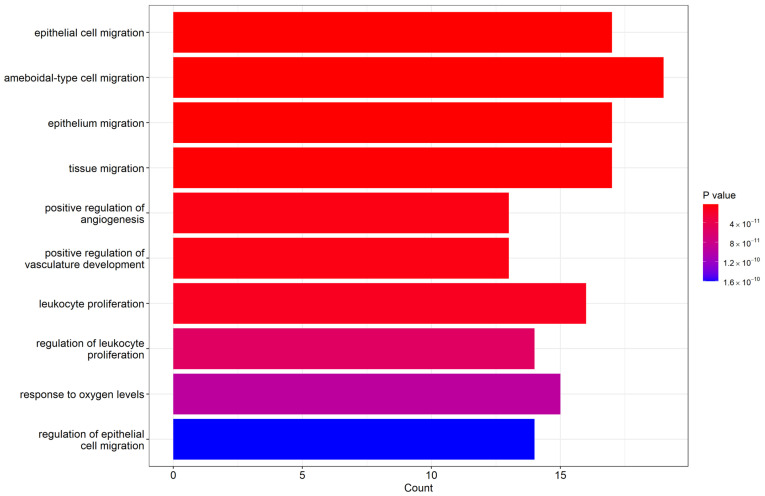
Top 10 biological process Enrichment Gene Ontology (GO) terms.

**Table 1 ijms-24-09748-t001:** Patient characteristics.

Case	Age	BMI	Cancer Stage	Cell Type	PCI Score	CC Score	NAC	HIPEC Regimen	PD	Current Status
A	66	23.98	4B	HGSC	11	0	yes	Carbo + Taxol	Yes *	Alive with disease
B	49	22.84	3C2	HGSC	6	0	yes	Carbo + Taxol	nil	NED
C	56	22.09	3C2	HGSC	16	0	yes	Carbo + Taxol	nil	NED
D	60	29.61	3C2	HGSC	13	1	nil	Carbo + Taxol	nil	NED
E	58	21.60	3C2	LGSC	5	0	nil	Carbo + Taxol	nil	NED
F	57	16.63	4A	HGSC	6	0	nil	nil	nil	NED

* Progression-free survivals: 8.57 months. BMI, body mass index; PCI, peritoneal carcinomatosis index; CC score, the completeness of cytoreduction score; HIPEC, hyperthermic intraperitoneal chemotherapy; PD, progression of disease; HGSC, high grade serous carcinoma; LGSC, low grade serous carcinoma; Carbo, carboplatin; Taxol, paclitaxel; NED, no evidence of disease. The Peritoneal Carcinomatosis Index (PCI) is a scoring system that gauges the distribution and severity of abdominal cancer, with a score range from 0, indicating no disease presence, to 39, denoting extensive disease, based on tumor size and its spread across 13 distinct peritoneal regions. The Completeness of Cytoreduction (CC) Score provides a rating of the efficacy of cytoreductive surgery, from CC-0, signifying the absence of visible residual disease, to CC-3, signifying the presence of non-resectable tumor nodules [28,29].

**Table 2 ijms-24-09748-t002:** miRNAs with significantly differential expression between pre-HIPEC and immediate post-HIPEC treatment (patient E).

	Pre-HIPEC	Immediate Post-HIPEC	ΔΔCq
hsa-miR-1254	2.27	7.19	4.91
hsa-miR-1290	7.36	10.22	2.85
hsa-miR-1972	13.43	17.19	3.76
hsa-miR-320a-3p	8.96	3.66	−5.31
hsa-miR-483-5p	2.14	4.88	2.73
hsa-miR-574-3p	2.06	8.32	6.26
hsa-miR-574-5p	10.09	12.36	2.27
hsa-miR-663a	11.06	5.22	−5.85

**Table 3 ijms-24-09748-t003:** miRNAs with significantly differential expression between CRS with and without HIPEC treatment.

	CRS + HIPEC (Patient E)	CRS Alone (Patient F)	ΔΔCq
hsa-miR-1290	10.30	7.92	2.39
hsa-miR-596	10.89	8.60	2.30
hsa-miR-1972	15.55	16.55	−1.01
hsa-miR-191-5p	7.90	2.77	5.13
hsa-miR-423-5p	6.15	10.30	−4.15
hsa-miR-320a-3p	7.53	5.78	1.75
hsa-miR-885-5p	7.59	1.07	6.52
hsa-miR-30b-5p	2.35	11.67	−9.32
hsa-miR-1228-5p	9.29	7.73	1.57
hsa-miR-589-5p	9.31	4.78	4.53
hsa-miR-202-3p	6.53	4.06	2.47
hsa-miR-151a-3p	9.31	10.51	−1.20
hsa-miR-378a-5p	5.97	8.24	−2.27

**Table 4 ijms-24-09748-t004:** Biological process gene ontology enrichment of miRNAs analysis.

GO	Description	Gene
GO:0010631	epithelial cell migration	SRF/MECP2/RAC1/HMGB1/ITGB3/NRP1/PTEN/KITLG/ANGPT2/FGF1/PIK3CA/TGFBR2/AKT3/IL4/KLF4/TGFB1/PIK3CD
GO:0001667	ameboidal-type cell migration	SRF/MECP2/RAC1/ACVR1B/HMGB1/AQP1/ITGB3/NRP1/PTEN/KITLG/ANGPT2/FGF1/PIK3CA/TGFBR2/AKT3/IL4/KLF4/TGFB1/PIK3CD
GO:0090132	epithelium migration	SRF/MECP2/RAC1/HMGB1/ITGB3/NRP1/PTEN/KITLG/ANGPT2/FGF1/PIK3CA/TGFBR2/AKT3/IL4/KLF4/TGFB1/PIK3CD
GO:0090130	tissue migration	SRF/MECP2/RAC1/HMGB1/ITGB3/NRP1/PTEN/KITLG/ANGPT2/FGF1/PIK3CA/TGFBR2/AKT3/IL4/KLF4/TGFB1/PIK3CD
GO:0045766	positive regulation of angiogenesis	ADM/HSPB6/AQP1/XBP1/ITGB3/NRP1/MTDH/ANGPT2/FGF1/TGFBR2/AKT3/KLF4/PIK3CD
GO:1904018	positive regulation of vasculature development	ADM/HSPB6/AQP1/XBP1/ITGB3/NRP1/MTDH/ANGPT2/FGF1/TGFBR2/AKT3/KLF4/PIK3CD
GO:0070661	leukocyte proliferation	MAPK3/MAPK1/IGF2/HMGB1/TFRC/BMI1/PTEN/KITLG/CRP/TGFBR2/IL4/CD274/JUNB/CEBPB/TP53/CDKN1A
GO:0070663	regulation of leukocyte proliferation	MAPK3/MAPK1/IGF2/HMGB1/TFRC/BMI1/PTEN/KITLG/CRP/TGFBR2/IL4/CD274/CEBPB/CDKN1A
GO:0070482	response to oxygen levels	SRF/MECP2/EP300/SMAD4/ADM/ND5/TFRC/AQP1/PTEN/ANGPT2/SIRT4/TGFBR2/JUND/SLC7A5/TP53
GO:0010632	regulation of epithelial cell migration	MECP2/RAC1/HMGB1/ITGB3/NRP1/PTEN/ANGPT2/FGF1/TGFBR2/AKT3/IL4/KLF4/TGFB1/PIK3CD

## Data Availability

Not applicable.

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
