# Peer review of "Peritoneal Fluid Analysis of Advanced Ovarian Cancers after Hyperthermic Intraperitoneal Chemotherapy"

_ijms, 2023, doi:10.3390/ijms24119748_

Round 1

Reviewer 1 Report

In the manuscript “Peritoneal fluid analysis of advanced ovarian cancers after hyperthermic intraperitoneal chemotherapy” the authors analyzed miRNA and cytokine expression changes in peritoneal fluid taken from patients undergoing HIPEC treatment during cytoreduction surgery, in order to evaluate the potential effect and response of HIPEC treatment. The in silico study is well conducted and statistically significant, but the authors do not attempt to validate in vitro any of the suggestive hypotheses identified by Pearson's correlation. The work has two important limitations, as underlined by the authors themselves, represented by the small number of patients and the time required to collect the samples. The discussion was good and captured the state of the art well. Thus, this article represents an elegant study that points to further studies to understand the potential of HIPEC treatment; in fact my last point refers to the potential of this data set, particularly if they are to be developed and expanded.

Reviewer 2 Report

Comments on ijms-2329709

Title: Peritoneal fluid analysis of advanced ovarian cancers after hyperthermic intraperitoneal chemotherapy

The study investigated miRNA and cytokine expression changes in peritoneal fluid samples of 6 patients with advanced ovarian cancers before and after receiving hyperthermic intraperitoneal chemotherapy (HIPEC) during cytoreduction surgery (CRS). Increased cytokine expression was observed, as well as the altered expression of several miRNAs. Correlative analyses were performed.

Major comments:

Introduction:

·    1.  Lines 74-87: Rewrite this paragraph to be more concise and informative about the role of miRNAs in cancer, and particularly ovarian cancer. Describe miRNAs that were identified as biomarkers in microarray studies from references 25,26. Describe mRNA targets of the miRNAs.

Results:

·        2. Figure 1 is confusing, because it is labelled 1-5, and your patients samples are also labelled 1-6, but you use patients 1-4 for step 4 of your study scheme, and patients 5-6 for step 5 of your study scheme. Either remove the labels from Figure 1, or label your samples in the text as A-F.

·      3.   Line 126: define abbreviation: Cq. Also, even after reading the methods section describing how this data was analysed, it is not clear how these values are calculated, why the number is significant if it is ≥1, what the range of numbers is, eg: what is the maximum number? Is 17.19 high on this scale? Is it a biologically significant level?

·    4.     Table 2: More information is needed in the text before table 2, describing what is being compared. Were the patient samples pooled for this analysis? Or are these values the averages from individual samples? If you have the data from individual samples, this should be given in supplementary data files.

·  5.       Section 2.3: Comparison between CRS+HIPEC vs CRS alone is 5 patients vs 1? This is what is stated in lines 107-108. This should be clearly stated in the text, in Table 3, and mentioned in the discussion as a limitation.

·  6.       Section 2.4: Again, are these the results of pooled samples or an average of individual samples?

·  7.       Table 4 is unnecessary, the data are given in Figures 3 and 4. Add the p-values to Figure 3 and remove table 4.

·  8.       Section 2.5: Why was this correlative analysis done? Briefly introduce the purpose for this analysis, indicating whether prior literature has reported correlations between miRNA expression and cytokine production.

·  9.       Figure 8 is unreadable, and uninformative. Indicate the miRNAs that are relevant to the pathways, what mRNAs they are regulating.

Discussion:

·    10.     The discussion is too long, and is mostly just a restatement of your results. Needs better interpretation of results and how the cited literature aligns with your findings. The discussion should be mostly around the findings with miR-1290, miR-320a-3p and miR-663a as they have the most interesting data. Suggest some future studies that would be done with additional sample collection.

·    11.     The effect of the sampling, range of timepoints and whether the samples were pooled for analysis needs to be discussed. These are limitations to the study that should be addressed.

·   12.      The cytokine expression changes should be discussed, in terms of the pattern that was similar for all except CCL2 and IL-6sR, peak and then decrease.

·    13.     The first paragraph of the discussion doesn’t flow and should be rewritten – HIPEC is introduced, and then the authors jump to cytokines and proteins that are increased in ovarian cancer ascites, with no link back to HIPEC, or statement of why the study was done.

·   14.      Lines 247-250 needs a citation, can’t just say “research has shown”.

·    15.     Lines 254-264, what is the relevance of these miRNAs to your findings? Comment whether these were detected in your study or remove this.

·    16.     Lines 286-304, this paragraph needs a summary of how the literature relates to your data.

·    17.     Lines 317-332, this paragraph could be cut almost completely, you only need to provide a link between your data and existing literature, not provide a literature review!

·   18.      Lines 333-366 again, restatement of results and a review of the literature, it is not a discussion.

·   19.      Lines 383-398 are well written like a discussion, this is how the rest of the discussion section should be.

Methods:

·   20.      Lines 506-511 should be included in the results section 2.2.

Minor Comments:

·     21.    Abbreviations should be corrected so that the full names are written out, followed by bracketed abbreviations. Eg: Interleukin-10 (IL-10).

·   22.      Lines 58-63: sentence doesn’t make sense, rewrite to clarify.

·     23.    Line 74: “associated miRNA” – associated to what? Also define miRNA abbreviation. Eg: micro RNA.

·    24.     Line 104: should be low-grade serous carcinoma

·  25.       Define PCI abbreviation in the text (line 106). Also define the PCI score range and meaning in Table 1.

·   26.      Define CC abbreviation in the text (line 106). Also define the CC score range and meaning in Table 1.

·  27.       Figure 3 and 6 legends should include more detail.

·  28.       Lines 208-212: sentence is too long, no need to list the top 10 GO terms in the text, just highlight major processes such as migration.

·  29.       Line 242: Akt is incorrectly described.

·  30.       Line 238 – transabdominal is misspelt.

·  31.       Lines 251-254 are unnecessary, this is the first and only time where exosomes are mentioned.

· 32.        Line 407 – “addition” should be “additional”

Round 2

Reviewer 2 Report

I thank the authors for responding to my comments and making changes to improve the paper. However, the study is not scientifically sound, basing the entire study from results on a comparison between 2 patients with very different disease as pre and post HIPEC comparisons. In your own words, “The study scheme is illustrated in Figure 1. The samples from cases 1, 2, 3, and 4 were collected and used for cytokine microarray evaluation. Also, samples from the other cases (5 and 6) were investigated for miRNA analysis and further ontology enrichment.” In other words, the comparison was made between case 5 and case 6.” So you described that only patients 5/E and 6/F were used for miRNA analysis, whereas more patients were used for cytokine analysis.  So this means that the data in Table 2 and Table 3 are presented from analysis of patient E and F only, and Figure 2 represents patient E only. It is misleading readers to call this comparison pre and post HIPEC, if you are comparing patient E and F here, as only patient E had HIPEC, and in fact has low grade serous cancer, whereas patient F did not have HIPEC at all according to Table 1, and is high grade serous cancer. You need to clearly describe exactly what patients and timepoints are being compared in Table 2 as this is the basis of the entire study. The study design is not logically presented, and seems more like an afterthought. I suggest the authors either do further analysis of their samples, at the very least to perform qRT-PCR on all patients to confirm detection of the miRNAs identified from the 2 patients used to justify the entire study, or submit to another journal, presenting this work as more of a case study or observational study.
